# PHYSICS-CONSTRAINED DEEPONET FOR SURROGATE CFD MODELS: A CURVED BACKWARD-FACING STEP CASE

**Anas Jnini**[1*], **Harshinee Goordoyal**[2*], **Sujal Dave** [3*], **Flavio Vella**[1], **Katharine H. Fraser**[2], and **Artem Korobenko**[3]

[1]University of Trento. {`anas.jnini, Flavio.vella`}@unitn.it
[2]University of Bath. {`hg757, K.H.Fraser`}@bath.ac.uk
[3]University of Calgary. {`sujal.dave, artem.korobenko`}@ucalgary.ca

## ABSTRACT

The Physics-Constrained DeepONet (PC-DeepONet), an architecture that incorporates fundamental physics knowledge into the data-driven DeepONet model, is presented in this study. This methodology is exemplified through surrogate modeling of fluid dynamics over a curved backward-facing step, a benchmark problem in computational fluid dynamics. The model was trained on computational fluid dynamics data generated for a range of parameterized geometries. The PC-DeepONet was able to learn the mapping from the parameters describing the geometry to the velocity and pressure fields. While the DeepONet is solely data-driven, the PC-DeepONet imposes the divergence constraint from the continuity equation onto the network. The PC-DeepONet demonstrates higher accuracy than the data-driven baseline, especially when trained on sparse data. Both models attain convergence with a small dataset of 50 samples and require only 50 iterations for convergence, highlighting the efficiency of neural operators in learning the dynamics governed by partial differential equations.

## 1 INTRODUCTION

Computational fluid dynamics (CFD) is commonly used for engineering design, in sectors ranging from automotive to aeronautics, biomedical engineering and energy production. CFD offers a more affordable option than experimental testing and allows for the evaluation of a wider range of designs from a fluid dynamic perspective. However, running high-accuracy simulations can be prohibitively expensive. Recently, various researchers have studied the use of data-driven and physics-informed machine learning to solve fluid dynamics problems.

Data-driven methods to create CFD surrogates include fourier neural operators (1), deep operator networks (2) and UNets (3). Lu et al. introduced the DeepONet, a deep neural operator framework, to learn linear and non-linear operators (function to function mapping) between small datasets (2). This framework is based on the universal approximation theorem for operators. Shukla et al. used DeepONets to create a surrogate for flow around airfoils which was significantly faster than traditional CFD methods . (4).

Physics-informed neural networks (PINNs), first introduced by Raissi et al. (5), aim to solve supervised learning tasks while respecting any given law of physics described by general nonlinear partial differential equations. They can be applied to fluid dynamics problems, as flows are modeled by the incompressible Navier-Stokes equations:

$$\nabla \cdot \mathbf{u} = 0 \tag{1}$$

---

[*]Equal contribution.

$$\rho \frac{\partial \mathbf{u}}{\partial t} + \rho(\mathbf{u} \cdot \nabla)\mathbf{u} = -\nabla p + \mu \cdot \nabla^2 \mathbf{u} + \mathbf{f}_b \tag{2}$$

where $\mathbf{u}$ is the continuum velocity, $p$ is the pressure, $\mu$ is the dynamic viscosity and $\mathbf{f}_b$ the resultant of the body forces (such as gravity and centrifugal forces) per unit volume acting on the continuum. PINNs have been used as surrogate models for CFD as they are able to predict flows for cases where little/no data is available (5; 6; 7). Moreover, solving the Navier-Stokes equations implies solving three sets of equations simultaneously: continuity (Eq 1), conservation of momentum (Eq 2) and boundary conditions. Several works have investigated embedding the boundary conditions or the divergence constraint imposed by the continuity equation directly by suitably modifying the network architecture, (8; 9; 10).

In this paper, the use of deep operator networks (DeepONet), proposed by Lu et al. (2), is explored to create a surrogate for predicting flow over curved backward-facing steps (BFS) with different parameterised curves.

Embedding fundamental principles, like the continuity equation, directly into the structure of neural networks can enhance the efficacy of cutting-edge neural operator models by narrowing down the potential solution space for our models (8). We propose a Physics-Constrained DeepONet (PC-DeepONet), an architecture that exactly satisfies the continuity equation (Eq 1). We show, through numerical experiments, that the proposed architecture increases the accuracy for the surrogate model, converging in only a few iterations with sparse training data.

We investigate the use of PC-DeepOnets through the lens of the backward-facing step (BFS). The BFS is a benchmark model that demonstrates important features such as flow separation, vortex evolution and re-attachment (11). The BFS has been studied by multiple researchers, especially in the fields of aerospace (12; 13; 14) and biomechanics (15; 16; 17). The curved BFS has been chosen for this study as in addition to displaying the same flow characteristics as the BFS, its separation region is sensitive to the boundary layer near the curved wall (18), making it suitable for learning the operator mapping between the shape of the curve and the resulting flow. Furthermore, curved surfaces are common in engineering applications, for example in aerodynamic bodies, blades, curved ducts and pipes, cylinders, and diffusers (18).

## 2 METHOD

In this study, a DeepONet is applied to learn the operator mapping between functions that parameterize the geometry of the BFS slopes and the flow field evaluated on the output grid. The DeepONet was chosen as it works well for learning functions between parameterised inputs and their corresponding outputs. An illustration of the geometry of choice can be found in Appendix A.1(Figure 2). The walls illustrated as solid lines were fixed while the curvature of the slope (dotted blue line) was varied.

The slope was parameterised using non-uniform rational B-spline (NURBS) which are used as mathematical representations of curves and surfaces. Using this method, curves can be parameterised using control points, weights, knot vector and degree of polynomial. Background information on NURBS can be found in (19). For the current study, the backward-facing curve is defined by a third degree (cubic) NURBS function with four control points, equal weights and a constant knot vector. The two control points at each end of the slope, marked as orange dots in Figure 1, are fixed. The other two control points, marked as green dots in Figure 1, are fixed along their respective y-axes and their x-coordinates are varied evenly between -0.25 and 1.25 to generate 50 different slope shapes. The x-coordinate of each control point is fed as one of the inputs to the network. As the y-coordinate is unchanged, there is no need to have this as an input.

### 2.1 DATA GENERATION

FreeFEM++ (20) was used to generate training data for flow over the curved BFS. FreeFEM++ uses the finite element method to solve the incompressible Navier-Stokes equations (Eq 1, 2) over the domain, by discretizing it into finite elements. For solution control, the incompressible flow solver combined a number of numerical techniques, such as an implicit scheme for time-stepping, finite

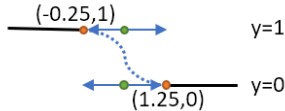

Figure 1: Control points for parameterisation of the BFS slope.

element spatial discretization, and iterative solvers for managing the resulting linear systems. To ensure accurate representation of physical phenomena, stabilization techniques such as the SUPG (Streamline Upwind/Petrov-Galerkin) method (21), which is particularly beneficial for handling convection-dominated flows, was employed. The simulations were run for 2D unstructured meshes, each comprising approximately 50000 elements, across 50 distinct geometries. Uniform boundary conditions were applied to all simulations. The flow was modeled as incompressible, with a fixed Reynolds number of 1000.

## 2.2 NETWORK ARCHITECTURE

The baseline DeepONet, as proposed by Lu et al. (2) consists of two sub-networks: a branch network and a trunk network as illustrated in Figure 3 in Appendix A.1. The input to the branch network consists of the x-coordinates of the control point pair for each geometry configuration. Hence, the size of the input to the branch net is the number of geometries x 2. The trunk net takes as input the output sensors: the spatial coordinates of the output mesh. The size of the input to the trunk net is therefore the number of sensors x dimensions. Three separate DeepONet networks with the same architecture were trained to predict the horizontal velocity (U), vertical velocity (V) and pressure(P) fields. For the PC-DeepONet, this architecture was extended by incorporating the approach proposed by Richter et al. (8) to make the velocity field divergence free .

The output of an individual DeepONet can be expressed as follows:

$$G^{\xi_q}(x,y) = \sum_{i=1}^{N_\alpha} \alpha_i(\xi_g; \theta_\alpha)\phi_i(x,y;\theta_\phi) \quad \forall q \in \{p,u,v\}. \tag{3}$$

Where $\phi$ a collection of basis functions that map functions from spatial coordinates to output functions, and $\alpha$ are a collection of coefficients that learn mappings from the parametrized slope control points to the output functions. Considering the two individual DeepONet output $b(x,y)$ as a vector field with components $b_u(x,y)$ and $b_v(x,y)$ corresponding to the velocities $u$ and $v$, we have:

$$b(x,y) = \begin{bmatrix} b_u(x,y) \\ b_v(x,y) \end{bmatrix} = \begin{bmatrix} G^{\xi_u}(x,y) \\ G^{\xi_v}(x,y) \end{bmatrix} = \begin{bmatrix} \sum_{i=1}^{N_\alpha} \alpha_i^u(\xi_g; \theta_\alpha)\phi_i^u(x,y;\theta_\phi) \\ \sum_{i=1}^{N_\alpha} \alpha_i^v(\xi_g; \theta_\alpha)\phi_i^v(x,y;\theta_\phi) \end{bmatrix}. \tag{4}$$

With this vector field, a skew-symmetric matrix $A$ is constructed from the Jacobian of $b$ and its transpose:

$$A = J_b - J_b^\top = \begin{bmatrix} 0 & \frac{\partial b_u}{\partial y} - \frac{\partial b_v}{\partial x} \\ \frac{\partial b_v}{\partial x} - \frac{\partial b_u}{\partial y} & 0 \end{bmatrix}, \tag{5}$$

where $J_b$ is the Jacobian matrix of $b$ with regards to the trunk input. The final divergence-free velocity field $v$ is represented by this divergence:

$$v = \mathrm{div}(A). \tag{6}$$

By construction, the divergence of $A$ is zero due to the properties of the skew-symmetric matrix, ensuring that the predicted velocity field is divergence-free, which aligns with the continuity condition.

## 2.3 TRAINING AND TESTING

Our model's implementation and optimization were conducted using the PyTorch framework. Training was executed on a dataset comprising 50 distinct geometries. For each geometry, we performed a mesh mapping to a target geometry to obtain a unique mesh using bi-cubic interpolation. 1000 output sensors were then randomly positioned across the mesh in points away from the slope to train the model. Since the mapping is done from function to function rather than from vector to vector, the model does not have to be trained on all the points in the domain and can learn the representation more efficiently with fewer sampled points. The DeepONet was then able to extrapolate for individual geometries, even in unseen domains. We illustrate this resolution-free capability of DeepONets by infering on a mesh with 50000 points, as shown in Figures 4 and 5 in Appendix A.1.

The optimization of the network parameters was performed using the Limited-memory Broyden-Fletcher-Goldfarb-Shanno (L-BFGS) algorithm. This particular optimizer was chosen to address vanishing gradient issues that arise when training DeepONets with hyperbolic tangents (tanh). Only 50 iterations were needed to achieve convergence for both architectures. The hyperparameters used for both DeepONet models are displayed in Table 1.

| Parameter | Value |
|---|---|
| Branch network architecture: | [2,40,40,40,100] |
| Trunk network architecture: | [2,40,40,40,100] |
| Activation functions: | tanh |
| $N_\alpha$: | 50 |
| Optimizer: | L-BFGS |

Table 1: Hyperparameters of the Optimized DeepONet

To evaluate the model's performance, we used the Mean Squared Error (MSE) loss function and assessed the relative $L_2$ error between the predicted and actual sensor data.

## 3 RESULTS AND DISCUSSION

We report the performance of the vanilla DeepONet and the PC-DeepONet models in Table 2. The PC-DeepONet, shows superior performance in both training and validation phases.

Both models were tested on unseen domains and geometries. The outputs from the trained models on a case from the validation set are displayed in Figure 4 and 5 in Appendix A.1.

Table 2: Performance comparison of DeepONet and PC-DeepONet.

| Model | Epoch | Training Loss | Validation Loss | Relative L2 Error |
|---|---|---|---|---|
| PC-DeepONet | 50 | $1.77 \times 10^{-6}$ | $1.82 \times 10^{-6}$ | $4.45 \times 10^{-3}$ |
| DeepONet | 50 | $1.24 \times 10^{-5}$ | $1.23 \times 10^{-5}$ | $1.16 \times 10^{-2}$ |

Both architectures achieved convergence within a limited number of iterations. The PC-DeepONet, with its enforced continuity condition, demonstrates superior accuracy with sparse data. We note that these results were obtained under conditions of smooth, laminar flow. We anticipate that for turbulent flows, the advantage of the PC-DeepONet will become more pronounced. Despite the resolution free-nature of neural operators, analysis of the Figures 4 and 5 reveals that errors cluster near boundary layers, likely due to under-sampling in the regions close to the slope, where points cannot be sampled in a common mesh. Future investigations will explore masking approaches to incorporate this information in the training and incorporating hard boundary conditions in the architecture.

## 4 Conclusion and Future Work

In this study, we introduce PC-DeepONets, an architecture that incorporates knowledge of the physics of fluid flow into the DeepONet to improve on data-driven learning. By enforcing the divergence constraint imposed by the continuity equation in the DeepONet framework, we observe significant improvements in accuracy for surrogate modeling with sparse data. Using PC-DeepONet, we successfully learned operational mappings between parameterized geometric configurations and flow field variables (U, V, and P).

Moving forward, integrating additional impositions by constraining boundaries and integrating additional information from the physics could potentially improve the performance of PC-DeepONets in capturing complex flow phenomena like turbulence for higher Reynolds numbers.

## Acknowledgements

We would like to thank the organisers of the 2023 Physics-Informed Neural Networks and Applications PhD Summer School at KTH for making this collaboration possible. We would also like to thank Dr Khemraj Shukla (Assistant Professor of Applied Mathematics (Research), Brown University) for his advice and support during this project.

AJ would like to acknowledge Leonardo S.p.A. (Italy). HG would like to acknowledge the ART-AI CDT (Bath, UK), funded by the UKRI. SD would like to acknowledge NSERC (Canada).

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

## A  APPENDIX

### A.1  ADDITIONAL MATERIAL

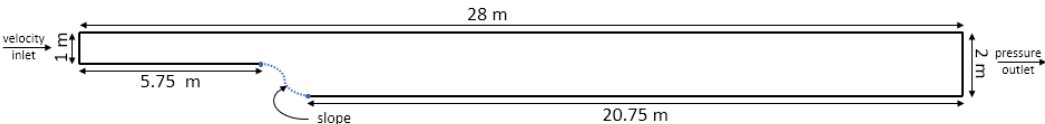

Figure 2: Illustration of the curved BFS used for training the DeepONet.

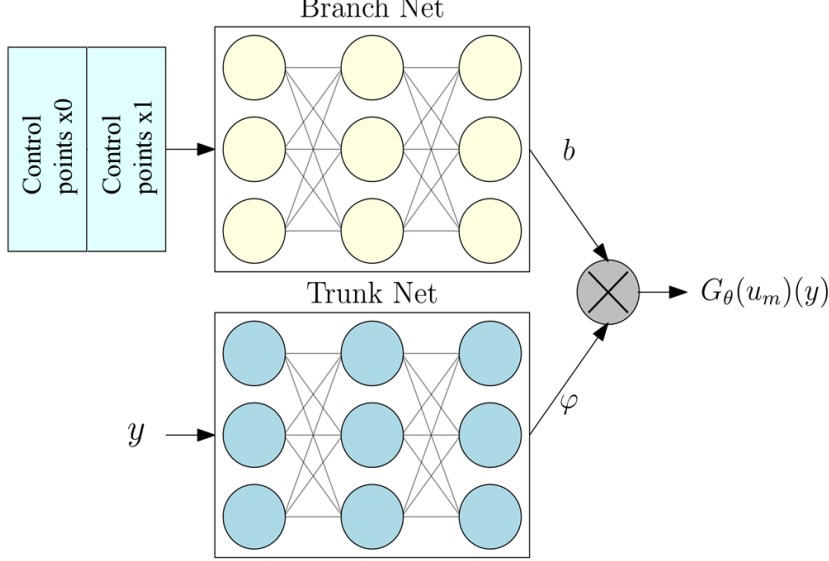

Figure 3: Architecture of the DeepONet used to learn the operator between slope shape and flow field (adapted from (22)).

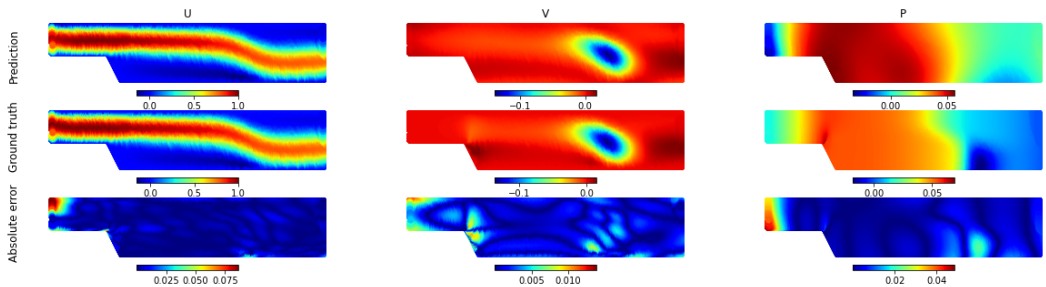

Figure 4: Comparison between the output of the DeepONet on an unseen test case with the ground truth for U, V and P.

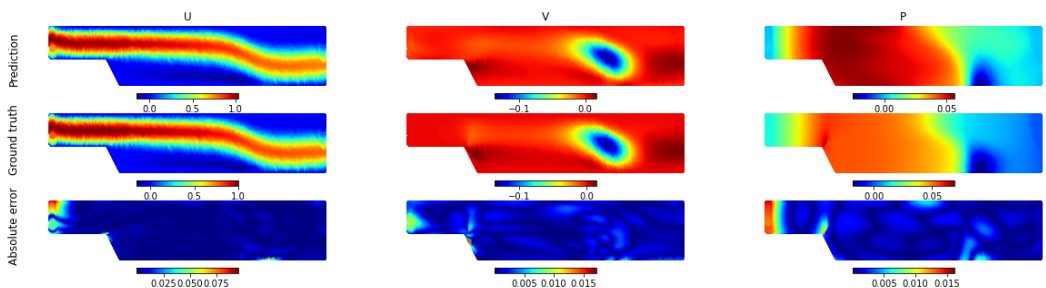

Figure 5: Comparison between the output of the Physics-Constrained DeepONet with the ground truth for U, V and P.

