# OpenReview forum: "Physics-constrained DeepONet for Surrogate CFD models: a curved backward-facing step case"
_ICLR.cc/2024/Workshop/AI4DiffEqtnsInSci — AI4DiffEqtnsInSci @ ICLR 2024 Poster_

### Official Review · Reviewer_kJVW · 2024-02-27
**A novel approach for using DeepONets with a convincing application**

**Rating:** 8
**Confidence:** 4

**Review:**

The paper is well motivated and showcases a convincing application of its proposed novel approach. It does lack a bit in terms of the details of the method, it would have been useful to get a deeper understanding of the way the physics constrained are introduced into the basis.

---

### Meta-Review · Program_Chairs · 2024-03-03

**Recommendation:** Accept (Poster)

**Metareview:**

The reviewer marks this as a clear accept and I agree. I encourage the authors to clarify the motivation of the physics constraints in the basis in the camera-ready version.

---

### Decision · Program_Chairs · 2024-03-03

Accept (Poster)